# Towards a Change in the Diagnostic Algorithm of Autism Spectrum Disorders: Evidence Supporting Whole Exome Sequencing as a First-Tier Test

**DOI:** 10.3390/genes12040560

**Published:** 2021-04-12

**Authors:** Ana Arteche-López, Maria José Gómez Rodríguez, Maria Teresa Sánchez Calvin, Juan Francisco Quesada-Espinosa, Jose Miguel Lezana Rosales, Carmen Palma Milla, Irene Gómez-Manjón, Irene Hidalgo Mayoral, Rubén Pérez de la Fuente, Arancha Díaz de Bustamante, María Teresa Darnaude, Belén Gil-Fournier, Soraya Ramiro León, Patricia Ramos Gómez, Olalla Sierra Tomillo, Alexandra Juárez Rufián, Maria Isabel Arranz Cano, Rebeca Villares Alonso, Pablo Morales-Pérez, Alejandro Segura-Tudela, Ana Camacho, Noemí Nuñez, Rogelio Simón, Marta Moreno-García, Maria Isabel Alvarez-Mora

**Affiliations:** 1Genetics Department, 12 de Octubre University Hospital, 28041 Madrid, Spain; mariajose.gomezr@salud.madrid.org (M.J.G.R.); mscalvin@salud.madrid.org (M.T.S.C.); juanf.quesada@salud.madrid.org (J.F.Q.-E.); josemiguel.lezana@salud.madrid.org (J.M.L.R.); carmen.palma@salud.madrid.org (C.P.M.); igomez@salud.madrid.org (I.G.-M.); irene.hidalgo@salud.madrid.org (I.H.M.); rpfuente@salud.madrid.org (R.P.d.l.F.); pramosg@salud.madrid.org (P.R.G.); olalla.sierra@salud.madrid.org (O.S.T.); alexandra.juarez@salud.madrid.org (A.J.R.); marranzcano@gmail.com (M.I.A.C.); m.moreno@salud.madrid.org (M.M.-G.); mialvarez@clinic.cat (M.I.A.-M.); 2Cancer Research Network (CIBERONC), 28029 Madrid, Spain; 3Genetics Department, Móstoles University Hospital, 28935 Madrid, Spain; adiazb@salud.madrid.org (A.D.d.B.); mariateresa.darnaude@salud.madrid.org (M.T.D.); 4Genetics Department, Getafe University Hospital, 28905 Madrid, Spain; belen.gilfournier@salud.madrid.org (B.G.-F.); soraya.ramiroleon@salud.madrid.org (S.R.L.); 5Neuropediatrics Unit, Móstoles University Hospital, 28935 Madrid, Spain; rebeca.villares@salud.madrid.org; 6Inmunology Department, 12 de Octubre University Hospital, 28041 Madrid, Spain; pmorales@h12o.es (P.M.-P.); alejandro.segura@salud.madrid.org (A.S.-T.); 7Child Neurology Unit, Neurology Department, 12 de Octubre University Hospital, 28041 Madrid, Spain; acamacho@salud.madrid.org (A.C.); noemi.nunez@salud.madrid.org (N.N.); rogeliosimon@gmail.com (R.S.); 8Biochemistry and Molecular Genetics Department, Hospital Clinic of Barcelona and Fundació Clínic per la Recerca Biomèdica, 08036 Barcelona, Spain

**Keywords:** autism spectrum disorder, diagnostic yield, exome sequencing, chromosomal microarray, *FMR1* testing, copy number variations

## Abstract

Autism spectrum disorder (ASD) is a prevalent and extremely heterogeneous neurodevelopmental disorder (NDD) with a strong genetic component. In recent years, the clinical relevance of de novo mutations to the aetiology of ASD has been demonstrated. Current guidelines recommend chromosomal microarray (CMA) and a *FMR1* testing as first-tier tests, but there is increasing evidence that support the use of NGS for the diagnosis of NDDs. Specifically in ASD, it has not been extensively evaluated and, thus, we performed and compared the clinical utility of CMA, *FMR1* testing, and/or whole exome sequencing (WES) in a cohort of 343 ASD patients. We achieved a global diagnostic rate of 12.8% (44/343), the majority of them being characterised by WES (33/44; 75%) compared to CMA (9/44; 20.4%) or *FMR1* testing (2/44; 4.5%). Taking into account the age at which genetic testing was carried out, we identified a causal genetic alteration in 22.5% (37/164) of patients over 5 years old, but only in 3.9% (7/179) of patients under this age. Our data evidence the higher diagnostic power of WES compared to CMA in the study of ASD and support the implementation of WES as a first-tier test for the genetic diagnosis of this disorder, when there is no suspicion of fragile X syndrome.

## 1. Introduction

Autism spectrum disorder (ASD) is one of the most prevalent groups of neurodevelopmental disorders (NDD), with a prevalence of one in 160 children worldwide [1] and a strong male bias (ratio ~4:1) [2]. According to the last version of the *Diagnostic and Statistical Manual of Mental Disorders* (DSM-5), it is characterised by early onset abnormalities in social and language communication and interactions, and by atypically restricted and repetitive behaviours and interests that can persist throughout life [3,4]. The phenotype of ASD is extremely heterogeneous, and patients often present other co-occurring clinical conditions including variable intellectual disability (ID) (~31%) [4], dysmorphic features, abnormalities in the electroencephalogram with or without epilepsy (~20–37%) [5], and/or other psychiatric and medical conditions such as anxiety or attention deficit hyperactivity disorders [6].

The aetiology of ASD is very heterogeneous. However, a strong genetic role is widely accepted [6,7] and thus, genetic testing is recommended. Fragile X syndrome (FXS) is considered the first monogenic familial cause of ASD and ID, representing 1–2% of ASD cases [8]. Copy number variants (CNVs) are observed in 5–10% of idiopathic ASD cases and are described in 8–21% of ASD individuals causing syndromic disease [9]. Therefore, current guidelines recommend chromosomal microarray (CMA) and *FMR1* testing for the diagnosis of FXS as first-tier tests for individuals with ASD, developmental delay/ID, and/or multiple congenital anomalies [10,11]. However, recent data show the majority of FXS cases have clinical features or family history suggestive of the disorder, proposing *FMR1* testing as a second-tier test for NDDs in the absence of FXS suspicion [12]. On the other hand, the use of next generation sequencing (NGS) has evidenced the contribution of de novo mutations to the aetiology of ASD [13]. Moreover, since the improvement of NGS in detecting large CNVs, recent data supports that NGS is more suitable for the diagnosis of NDDs compared to CMA [14,15,16]. Specifically in ASD, the reported diagnostic yield by NGS ranges from 3 to 28%—depending on the complexity of the ASD phenotype—[17,18] whereas it ranges from 7 to 9% by CMA [19].

Here, we retrospectively study the genetic diagnostic yield in 343 ASD patients with and without other co-occurring clinical conditions by using different genetic approaches (*FMR1* testing, CMA, and/or whole exome sequencing (WES)). We extensively describe and compare the diagnosis and clinical utility of WES, CMA, and *FMR1* testing and describe the causal identified genes in a large cohort of ASD patients. We also highlight the importance of considering the phenotype and the age of the patient when ordering genetic testing. Our results support the implementation of WES as the first-tier approach in the diagnosis of ASD patients, when there is no suspicion of FXS.

## 2. Materials and Methods

### 2.1. Patients and Samples

All patients included in this study were referred for genetic testing to the Genetic Service of the Hospital Universitario 12 de Octubre from January 2017 to November 2020 with the clinical suspicion of ASD. Depending on the age of the patient and prior the analysis, all patients underwent clinical examination by the Neurology or Neuropediatry Department of the Hospital Universitario 12 de Octubre, the Hospital Universitario de Getafe or the Hospital Universitario de Mostoles. Likewise, all patients received adequate pre- and post-test genetic counselling.

ASD diagnosis was performed according to DSM-5 clinical criteria. All the patients showed persistent deficits in social communication and interaction as well as restricted, repetitive behaviours. Data on sex, age, and available previous genetic testing were retrospectively collected. A total of 343 patients were analysed, including 258 males (75.2%) and 85 females (24.3%) with a mean of age (±SD) at diagnosis of 6.0 ± 5.4 years, ranging from 1 to 45 years old. Heterogeneous co-occurring conditions were referred in ~40% of patients (137/343), being the presence of developmental delay/ID the most frequent (57%; 78/137). Other less frequent clinical co-occurring conditions included attention deficit hyperactivity, epilepsy, micro/macrocephaly, and the presence of any dysmorphic features/malformations. For analytic purposes, patients were classified in two groups according to the age of the patient at which the analysis was requested (before or after 5 years old) (Table 1). This threshold was chosen because at that age a clinical diagnosis of cognitive impairment can be established.

From the 343 patients, there were (i) 106 patients in which *FMR1* testing, CMA and WES were performed (FMR1 + CMA + WES); (ii) 83 patients studied by CMA and WES (CMA + WES); (iii) 118 patients with a CMA of whom 72 had also a *FMR1* testing performed (CMA ± FMR1) and iv); 36 patients with a WES performed, of whom 27 were also studied for FXS (WES ± FMR1) (Table 1).

From January 2017 until September 2018, the algorithm used for the genetic diagnosis of ASD patients included a CMA analysis for all cases together with a *FMR1* testing in the majority of cases (CMA + FMR1). In October 2018, WES was implemented in the Genetic Service of the Hospital Universitario 12 de Octubre including both single nucleotide variants (SNVs) and CNV detection. Since then, WES was performed to ASD patients with a previous negative genetic test (FMR1, CMA, or CMA+FMR1). In some patients, especially in those with other co-occurring clinical conditions, WES was firstly performed.

All patients or their guardians provided written informed consent for testing and for the use of their clinical and genetic data. The study was approved by the Ethics Committee of the University Hospital 12 Octubre. The study was carried out in accordance with the Declaration of Helsinki (2013).

Whole peripheral blood samples from *probands* and all available relatives were collected in EDTA tubes. Genomic DNA extraction was further performed, following standard procedures.

### 2.2. Molecular Genetics

#### 2.2.1. Chromosomal Microarray

Among the total 307 CMA, 173 (56.3%) were outsourced and performed by using a 180 K CMA (180 K KaryoNIM^®^, NIMGenetics^,^ Madrid, Spain). This platform is specifically designed for ASD patients (Agilent Tech, Santa Clara, CA, USA) with a resolution of 15 kb for the currently known 140 ASD-related genes and 100 kb for the rest of the genome. In 69 patients (22.5%), a 60K CMA was performed in our laboratory. This platform provides a resolution of 350 kb in the backbone and of 100 kb in targeted regions (60 K KaryoNIM^®^, NIMGenetics, Madrid, Spain). Standard procedures were followed in both approaches. The hg19 and the ADM-2 algorithm (Aberration Detection Method-2) were used. Analysis and interpretation were performed using Cytogenomics (v.4.0.3.12, Agilent) software. A threshold of ≥5 consecutive probes was established to consider a CNV.

No information regarding the specific outsourced platform used was available in the remaining 65 patients (21.2%).

Segregation analysis was performed when applicable by using a 60 K or 180 K CMA, following manufacturer’s instructions. A karyotype was performed in one patient to confirm the CMA results.

CNVs detected were classified following previous CMA recommendations for clinical practice [20,21]. “Characterised cases” were defined as those with pathogenic (P) or likely pathogenic (LP) CNV associated with phenotype. “Indeterminate cases” were defined as those patients with a CNV whose pathogenicity or contribution to disease was not totally certain. “Unsolved or negative cases” included those with polymorphic and/or no reported CNVs.

#### 2.2.2. Whole Exome Sequencing

WES was performed in our laboratory using the kit xGen Exome Panel v1.0 (IDT—Integrated DNA Technologies, San Diego, NJ, USA). Paired-end sequencing (2 × 75 bp) was carried out on a NextSeq 550 (Illumina) and bioinformatics analysis was conducted using a custom pipeline (Karma) that followed the recommendation of the Association for Molecular Pathology and the College of American Pathologists [22]. Reads were aligned to the reference human genome (hg19) using BWA MEM (v0.7.17) and Bowtie2 (v.2.4.1) The variant calling process was performed using GATK (Haplotype Caller from Genome Analysis Toolkit, v.4.1) and VarDict (AstraZeneca, v1.7.0) [23]. Annovar (v2018Apr16) [24] was used for the annotation of variants. ExomeDepth R package (v1.10) was used for CNV identification and AnnotSV (v2.4) for CNV annotation.

Variants that did not fulfil the established quality criteria were filtered out. In addition, those with a frequency ≥3% in gnomAD population database (v2.1.1), variants classified as benign or probably benign by multiple subscribers in ClinVar database (March 2020 release), synonymous variants within ±20 bp from the canonical splicing site, and intronic variants localised more than 15 nucleotides from the exon/intron junction were also filtered out. Sequence variants and copy number variants were prioritised.

Data analysis was based on a custom panel that included 293 genes related to ASD. Variant filtering was conducted according to quality parameters, variant type, pathogenicity predictor scores, and variant frequencies in population control databases such as allelic frequencies in Genome Aggregation Database (v2.1.1) and frequencies in our in-house database of variants (12OVar).

Variants were classified following the American College of Medical Genetics (ACMG) criteria [25]. “Characterised cases” were defined as the presence of one or two disease-causing variants (P or LP). “Indeterminate cases” included those cases with at least one variant of uncertain significance (VUS) in which additional required studies to confirm/discard their possible clinical implications were unavailable. “Negative cases” were defined as those in which no relevant disease-causing variants were reported.

Segregation analysis of detected disease-related variants and CNVs was performed by Sanger sequencing or either a multiplex ligation-dependent probe amplification (MLPA) or a 60 k CMA, depending on the availability of the MLPA technique and the size of the CNV.

#### 2.2.3. FMR1 Testing

Patients derived from the Neuropediatry Service of the Hospital 12 de Octubre were referred to the Immunology Service for *FMR1* testing using the Amplidex kit PCR/CE FMR1 (Asuragen, TX, USA) following manufacturer’s recommendations. In patients derived from the Hospital Universitario de Mostoles or Hospital Universitario de Getafe, the *FMR1* testing was outsourced to external/private laboratories.

### 2.3. Statistical Data Analysis

Statistical analyses were performed using commercially available software (SPSS-PC, version 23.0; SPSS Inc., Chicago, IL, USA). The Chi-square test of frequencies and Fisher’s exact test were applied to the contingency tables. Significance was accepted for exact asymptotic bilateral *p*-values below 0.05.

## 3. Results

Genetic testing was performed in 343 patients with a clinical indication of ASD, including *FMR1* testing, CMA, and/or WES. The number of characterised, indeterminate, and negative ASD cases according to the age of the patients at diagnosis is summarised in Table 2. The total genetic approach performed in each group is indicated. Only characterised patients were considered to determinate the diagnostic yield.

We have identified causative variants in 44 patients (12.8%) of our ASD cohort. Among them, WES identified 75% of causative variants (33/44) whereas CMA and *FMR1* testing respectively identified 20.4% (9/44) and 4.5% (2/44). In addition, VUS were observed in 61 patients (17.8%), 34 of them detected by CMA and 26 detected by WES.

Statistical analysis found significant differences when comparing overall diagnostic yields by WES (14.7%; 33/225) to either CMA (2.9%; 9/307) or *FMR1* testing (0.9%; 2/206) (*p* < 0.001).

In order to assess the diagnostic yield of patients diagnosed when less than 5 years old from those diagnosed when over 5 years old, the percentages of characterised cases were determined in each group. Statistically significant differences were observed between groups (*p* < 0.001). While 22.5% (37/164) of patients over 5 years old were characterised, only 3.9% (7/179) of patients under 5 years old were found to carry a causative genetic alteration. Indeed, among the characterised patients, 84% (37/44) were older than 5 years old, whereas 15.9% (7/44) were diagnosed when less than 5 years old. 

### 3.1. Patients Characterised by CMA

From the 307 patients in whom a CMA was performed, pathogenic CNVs were found in 9 cases (~3%) including (i) 6 deletions that consisted of a 3.76 Mb deletion in 8p23.13 associated to 8p23.13 deletion syndrome (ORPHA251071), a 0.59 Mb deletion affecting the *RBFOX1* gene, a 0.02 Mb deletion encompassing part of the *MBD5* gene associated to autosomal dominant mental retardation (#156200), a 2.98 Mb deletion in 22q11.21 associated with DiGeorge syndrome (#188400), and two deletions of 1.7 and 0.4 Mb size in chromosome 15 that respectively correspond to 15q13 (#612001) and 15q11 (#615655) microdeletion syndrome; (ii) 2 duplications of 6.19 and 4.78 Mb in 15q11.2q13.1 and 1q21.1–q21.2 that respectively were associated with 15q11–q13 (#608636) and 1q21 duplication syndrome (#612475); and (iii) a duplication of the entire chromosome Y for which a 47,XYY[121]/46,XY[32] mosaic was further confirmed by karyotype (Table 3).

The two largest CNVs were of de novo origin whereas the deletion of 0.59 Mb size involving the *RBFOX1* gene, the deletions in 15q13.2 and 15q11.2 of respectively 1.7 and 0.4 Mb size, and the duplication of 4.78 Mb size in 1q21.1 were inherited from an apparently asymptomatic parent. All of them presented with incomplete penetrance and variable expression. No information regarding the origin of the deletion of 0.02 Mb size encompassing part of the gen *MBD5* neither the origin of the 22q11.21 deletion was available (Table 3).

All patients characterised by CMA were males with the exception of one female that was diagnosed with DiGeorge syndrome. All males carrying a pathogenic deletion presented with psychomotor delay; one of them also had congenital heart disease and dysmorphic features. The patient with the 15q11–q13 duplication syndrome was an autistic patient that also manifested with epilepsy. Finally, the 47,XYY[121]/46,XY[32] mosaic ASD patient was referred with language difficulties (Table 3).

Among patients analysed by CMA, we found 11% (34/307) of indeterminate cases in whom a definitive diagnosis could not be established. Specifically, we detected 36 CNVs classified as VUS (Appendix A).

### 3.2. Patients Characterised by WES

From the 225 patients in whom a WES was performed, pathogenic or likely pathogenic variants were found in 33 cases (14.6%) (Table 4).

Segregation analysis revealed that approximately 75% (25/33) of the cases were associated to de novo variation, including four cases in which the presence of the variant could only be discarded in one progenitor. Remarkably, a de novo deletion of 1.89 Mb size involving the *NAGA*, *CYB5R3*, and *TCF20* genes (chr22:42,264,616–44,068,185) was detected by WES and further confirmed by 60K CMA.

Interestingly, in the genes *ANKRD11*, *MECP2*, and *MED13L*, we found two unrelated patients harbouring a de novo causing variant: (i) 2 loss-of-function (LOF) variants p.(Thr2362Profs*38) and p.(Tyr2469*) in the *ANKRD11* gene in a female and a male, both older than 4 years old and presenting with developmental delay and peculiar facial features; (ii) 2 causative variants p.(Arg20Glufs*30) and p.(Arg145Cys) in the *MECP2* gene (NM_00111079) in two females referred with ASD, psychomotor retardation, and seizures; and (iii) 2 de novo LOF variants c.1280+1G>T and p.(Q704*) in the *MED13L* gene (NM_015335.4) in a 14-year-old patient referred with ASD features and in 11-year-old patient with ASD traits, absence of language, cutis marmorata, and hypospadias (Table 4).

In our cohort, 9% of patients (3/33) carried biallelic variants in genes associated to neurodevelopmental disorders with autosomic recessive inheritance. Specifically, biallelic mutations were detected (i) in the *PIGG* gene in an 11-year-old male with ASD Asperger-type; (ii) in the *PMM2* gene in a 14-year-old female with ASD, hearing loss, and ptosis; and (iii) in the *CEP290* gene in a 13-year-old male with ID and attention deficit (Table 4).

In addition, in 2 cases (2/33) the following likely pathogenic causative variants were also inherited: (i) a variant maternally inherited in the *FGD1* gene associated to recessive X-linked syndromic mental retardation 16 (#305400) and (ii) a frameshift variant in the *CLTC* gene inherited from the apparently unaffected mother associated to autosomal dominant mental retardation 56 (#617854), a disorder with highly variable severity. In the remaining 3 cases (3/33), segregation analysis of the variants detected in the genes *TCF4*, *IFIH1*, and *SHANK3* could not be performed but as they were classified as pathogenic, cases were considered to be characterised (Table 4).

Among the 33 patients characterised by WES, the vast majority of patients were more than 5 years old at the time of diagnosis (28/33; 84.8%). Only 5 patients under 5 years old were characterised by WES (5/33; 15.15%).

Finally, WES identified 44 VUS in 26 patients (26/224; 11.6%). Segregation analysis or functional data were not available for these variants. Intriguingly, in two unrelated patients, two different missense (classified as VUS without segregation analysis) were detected in the *MED13* gene, a gene associated to intellectual developmental disorder (#618009) (Appendix A).

### 3.3. FMR1 Testing

*FMR1* testing was performed to 206 patients as a first-tier diagnostic test. Patients not screened consisted of patients with no clinical manifestations suggestive of *FMR1*-related pathologies or in whom this result could not be collected. Among the 206 patients analysed, all patients carried normal alleles (<45 repeats) with the exception of 2 males (0.97%; 2/206) that were carriers of premutation alleles (61 and 72 CGG repeats). Additional genetic alterations were discarded by CMA and WES in both patients, supporting the involvement of the premutation in the clinical manifestations of these patients.

## 4. Discussion

It has been more than ten years since current guidelines about genetic testing in NDDs were reported [11]. The introduction of NGS technologies has revolutionised the field of genetic diagnosis and there is an ongoing debate about whether high-throughput technologies should be performed as a first-tier test for the diagnosis of NDDs, especially since NGS has overcome the limitations of CNV detection. Accumulating evidence suggests that NGS technologies are more promising compared to CMA [14,15,16,26]; however, they have not been extensively evaluated in ASD to date [17,18]. Regarding the *FMR1* gene, the *FMR1* expansion is considered the most common form of monogenic ASD [27] and is currently included in the algorithms for studying NDDs. Nevertheless, either full mutation or premutation alleles only explain 2–4% of ASD cases [28,29], and recent data show the majority of FXS cases have clinical features or family history suggestive of the disorder [12]. We report the diagnostic yield comparisons obtained in 343 ASD patients by using different genetic approaches (*FMR1* testing, CMA, and WES).

Our report supports that WES should be considered as first-tier test in the genetic diagnosis of ASD since a greater diagnostic yield was statistically significant when compared to either CMA or *FMR1* testing. Specifically, WES identified 14.6% of causative variants in ASD patients whereas diagnostic yield of CMA and *FMR1* testing was 2.9% and 0.9%, respectively. In fact, 75% (33/44) of the characterised patients were diagnosed by WES, demonstrating the superiority of WES over CMA and *FMR1* testing. Accordingly, a recent meta-analysis has analysed the genetic diagnostic yields of WES and CMA in patients with global developmental delay, ID, and/or ASD [26]. Based on this analysis, the authors propose a diagnostic algorithm placing WES at the beginning for the evaluation of unexplained NDDs. If no genetic alteration is observed and CNV detection is not available, they recommend CMA as the second genetic test [26].

In this report, we overall characterise ~13% of our cohort (44/343), since not all cases were tested by the three genetic approaches (Table 1), a percentage similar to the diagnostic yield reported in other ASD series [17]. Most related-ASD genes interact within the context of gene networks, synaptic function, and signalling pathways [30], and de novo mutations have been described to be strongly associated with autism [13,31,32] Accordingly, the vast majority of the genetic alterations in our cohort had a de novo origin, specifically 2 CNVs and 25 SNVs from the total 44 characterised cases. The majority of the identified causal genes was unique and had been previously described in the Simons Foundation Autism Research Initiative (SFARI) database associated with ASD, with the exception of 7 genes (*CLTC*, *FGD1*, *HIST1H1E*, *IFIH1*, *PIGG*, *PMM2*, and *SOX11*) associated with other disorders in which ASD-like symptoms could also be present. Only three genes (*ANKRD11*, *MECP2*, and *MED13L*) were found to be altered twice in non-related patients. Interestingly, ~7% (3/44) of characterised ASD patients had biallelic disruptive variants in *CEP290*, *PMM2*, and *PYGG* genes associated with an autosomal recessive pattern of inheritance. In concordance, Doan et al. (2019) have demonstrated that approximately 5% of ASD cases correspond to recessive gene disruptions [33] In addition, 2 cases with SNVs associated to X-linked disorders and 4 inherited CNV with incomplete penetrance were found. Finally, we also identified a male with a Y-chromosome aneuploidy (47,XYY[121]/46,XY[32] mosaic), which has been described to be 20 times more likely in ASD males than in the general population [34]. Segregation was not available in five cases (2 CNV and 3 SNVs) nor in the 2 premutation males.

The introduction of WES as first-tier test might provide several advantages without implying a significant increase in the turnaround time and costs when compared to CMA. Firstly, it might lead to the detection of the SNV variants in monogenic forms of ASD that correspond to 75% (33/44) of the genetic alterations detected in our studied cohort. Secondly, improvements of bioinformatics pipelines have led to the accurate detection of CNVs by NGS data as reflected by the identification of a de novo 1.89 Mb CNV that was further confirmed by CMA. Indeed, it is likely that all the 9 CNVs detected by CMA, even the 47,XYY[121]/46,XY[32] mosaicism, would have also been detected by WES in optimal conditions, since all CNVs encompassed at least one gene included in the custom ASD panel and enough probes to possibly detect them by WES. Therefore, the sensibility of the diagnosis would not have changed if WES had been used in the first place. However, this technology is not the gold standard for CNV detection [21] and it is currently recommended to validate this type of genetic alteration by other technology such as CMA or MLPA. Finally, WES offers the possibility to reanalyse negative cases adding new genes, which are continually increasing in ASD [35].It has been reported that NGS reanalysis may increase the diagnosis yield up to ~30%, albeit that this rate is not specific for ASD [36,37].

On the other hand, the broad phenotypic spectrum of ASD makes even more challenging to reach a genetic diagnosis. The term *autism spectrum disorder* includes a wide range of clinical manifestations that varies in the type and severity of symptoms, ranging from patients fully able to perform all daily activities to others requiring substantial support for basic activities. In the literature, a higher diagnostic yield (up to 30%) has been reported in ASD patients presenting with other clinical features compared to those who manifest an “isolated” ASD form (3%) [17,18]. We observed that the majority of the characterised patients in our cohort corresponded to ASD cases in which other co-occurring conditions were also referred, being psychomotor delay and/or ID the most frequent (in 69% of patients (29/42; Table 3 and Table 4). As previously reported by Tammimies et al. 2015, it is reasonable that the diagnostic rate in these patients is closer to the 30–50% reported in the literature [26,38,39] and to the 35–40% obtained in our laboratory for NDDs (data not shown). Recently, a clinical correlation between CNVs and ASD has been suggested [40]. The authors showed that the presence of dimorphisms and microcephaly is significantly overrepresented in patients with causative CNVs. Interestingly, we find that the presence of dysmorphic features/ peculiar phenotype was more frequent in patients with causal CNVs (5/10; 44%) than in patients with causal SNVs/indels (9/32; 28%) (Table 3 and Table 4).

The age at which patients should be offered genetic testing is also controversial. Interestingly, 84% (37/44) of characterised patients were 5 years old or more. Overall, the genetic diagnostic yield in the younger patients was ~4% (7/179). Only 7 patients under 5 years old were characterised, suggesting the genetic analysis in these patients is a challenge.

Regardless the genetic approach, approximately ~70% (238/343) of our ASD patients remain undiagnosed since no candidate genetic alterations were detected. This large percentage might be explained by the multi-factorial origin of ASD as both common and rare genetic variants contribute to autism risk [41]. A recent study has defined 102 high confidence genes, of which 53 seem to have more influence on social behaviour [42]. However, insufficient evidence to determinate “autism-specific” genes based on large-effect rare-variant has been recently suggested [43] and it is likely that there is no single cause (genetic, environmental or cognitive) defining autism [44]. Even when a variant is identified, other multiple rare and common genetic variants contribute to the psychiatric traits in ASD patients and, thus, to the clinical and genetic heterogeneity of the disorder. It might be interesting to reanalyse the indeterminate cases (61/343) with VUS in order to shed light on this issue.

We are aware about the limitations of our study, including the limited sample size and the lack of functional/segregation analysis in indeterminate cases and the absence of a detailed cost-effectiveness assessment.

ASD is highly heritable but clinical and genetically heterogeneous with both common and rare genetic variants collaborating to predispose individuals to the disorder. The discovery of new related genes has clarified the genetic architecture of ASD. Our study provides further evidence supporting the implementation of WES as the first-tier test for the diagnostic of ASD patients. Genetic diagnosis has a direct benefit not only for the clinical management of the patient but also in their relatives, which can benefit from genetic counselling and prenatal diagnosis. The current challenge is to translate the better knowledge of the molecular basis of ASD into an understanding of pathological mechanisms, as a step toward the development of more effective treatments.

## Figures and Tables

**Table 1 genes-12-00560-t001:** Demographics and genetic tests performed in each group of autism spectrum disorder (ASD) patients.

	Patients < 5 Years Old (N = 179)	Patients ≥ 5 Years Old (N = 164)	Total (N = 343)
**Gender (male/female)**	133/46	125/39	258/85
**Age at diagnosis, mean (min–max)**	2.82 (1–4)	9.46 (5–45)	6.0 (1–45)
**Genetic testing, n**			
**FMR1 + CMA + WES**	53	53	106 (30.9%)
**CMA + WES**	37	46	83 (24.2%)
**CMA (±FMR1)**	65	53	118 (34.4%)
**WES (±FMR1)**	24	12	36 (10.5%)
**Total**	179	164	343 (100%)

FMR1 = *FMR1* testing, CMA = chromosomal microarray; WES = whole exome sequencing.

**Table 2 genes-12-00560-t002:** Main results of the genetic test performed. Characterised, indeterminate, and negative ASD cases according to the age of the patient at which the analysis was requested are shown. Percentages are calculated from the total row. CMA = chromosomal microarray; WES = whole exome sequencing.

	Characterised Cases	Indeterminate Cases	Negative Cases	Total
**Patients < 5 years old**	7 (3.9%)	28 (15.7%)	144 (80.4%)	179 (100%)
**WES**	5/114	13/114	96/114	
**CMA**	2/155	17/155	136/155	
***FMR1* testing**	0/116	0/116	116/116	
**Patients ≥ 5 years old**	37 (22.5%)	31 (19%)	96 (58.5%)	164 (100%)
**WES**	28/111	12/111	71/111	
**CMA**	7/152	19/152	126/152	
***FMR1* testing**	2/90	0/90	88/90	
**Total general**	44 (12.8%)	59 (17.2%)	240 (70%)	343 (100%)

**Table 3 genes-12-00560-t003:** Characteristics of cases identified by chromosomal microarray (CMA)

ID	Sex	Age of Diagnosis (Years)	Clinical Indication for the Study	CMA Platform	CMA Results (Coordinates)	Size (Mb)	Genetic Diagnosis (OMIM # if Available)	Included OMIM Genes	Origin
**19AC29**	M	6	Attention deficit, developmental delay, congenital heart disease, and dysmorphic features	60 K	8p23.13.4 (8,100,384–11,860,230)X1	3.76	8p23.1 deletion syndrome	21 genes	Dn
**17AC33**	M	8	ASD and psychomotor delay	180 K	16p13.3 (5,874,625–6,466,890)X1	0.59		*RBFOX1* (exons 1–2)	Pat asym
**19AC272**	M	13	ASD and psychomotor delay	180 K	2q23.1 (149,135,883–149,154,803)X1	0.020	2q23.1 syndrome or autosomal dominant mental retardation (MIM #156200)	*MBD5* (intron 5–6)	n/a
**5867**	M	0.8	ASD and psychomotor delay	180 K	15q13.2 (30921917–32618383) X1	1.7	Microdeletion 15q13 syndrome (MIM #612001)	7 genes: *ARHGAP11A*, *FAN1*, *MTMR10*, *TRPM1*, *KLF13*, *0TUD7A*, and *CHRNA7*	Mat asym
**6210**	M	5	ASD	180 K	15q11.2 (22759178–23155311) X1	0.4	Microdeletion 15q11 syndrome MIM #615656)	4 genes: *TUBGCP5*, *CYFIP1*, *NIPA1*, and *NIPA2*	Pat asym
**17AC114**	M	5	ASD and epilepsy	180 K	15q11.2q13.1 (22,668,852–28,859,449) X3	6.19	Duplication syndrome 15q11–q13 (MIM #608636)	23 genes	Dn
**5700**	M	7	ASD and dysmorphic features	180 K	1q21.1–q21.2 (144,895,322–149,680,340) X3	4.78	1q21 Duplication syndrome (MIM#612475)	33 genes	Pat asym
**6356**	F	10	Intellectual disability, ASD, and dysmorphic features	180 K	22q11.21 (18,729,744-21,705,113) X1	2.98	DiGeorge syndrome (MIM #188400)	45 genes	n/a
**20AC133**	M	1	ASD	60 K	Yp11.32p11.2 (2,184,259–10,029,472) X2; Yq11.21q12 (13,675,923–28,804,541) X2	7.8 15	47,XYY/16,XY syndrome		n/a

ASD = autism spectrum disorder; CMA = chromosomal microarray;Dn = de novo; ID: identification, Pat asym = asymptomatic father; n/a = not available. M = male; F = female.

**Table 4 genes-12-00560-t004:** Characteristics of ASD cases identified by WES.

ID	Sex	Age of Diagnosis (Years)	Clinical Indication for the Study	CMA	CMA Results	Gene	Gene NM_	Cigosity	Coding Change	Protein Change	Origin
**Patients under 5 years old**											
18AC0044	F	1	Psychomotor delay, ASD, and epilepsy.	180 K	2q23.1del mat	*MECP2*	NM_001110792.1	het	c.433C>T	p.Arg145Cys	Dn
20NG0674	M	2	Psychomotor delay and language delay.	≥60 K	Normal	*FGD1*	NM_004463.2	hem	c.1327C>T	p.Arg443Cys	Mat asym
20NG0517	M	1	Psychomotor retardation delay, and behavioural problems. Rough face.	60 K	8p23.1dup	*IQSEC2*	NM_001111125.2	hem	c.2278G>A	p.Gly760Ser	Dn
20NG0002	M	2	Psychomotor and language delay. Sotos-like conduct disorder.	60 K	Normal	*NSD1*	NM_022455.4	het	c.1953del	p.Ile652Ter	Dn
19NG0815	M	3	Psychomotor delay and ASD.	≥60 K	15q13.2del, 15q13.3dup	*CUL3*	NM_003590.4	het	c.802delC	p.L268Sfs*5	No pat
**Patients of 5 years-old or older**											
19NG0707	M	11	No language. ASD traits. Cutis marmorata and hypospadias.	≥60 K	Normal	*MED13L*	NM_015335.4	het	c.2110C>T	p.Gln704Te	Dn
19NG0592	M	14	ASD features.	n/a	n/a	*MED13L*	NM_015335.4	het	c.1280+1G>T	p.?	Dn
20NG0744	F	19	Psychomotor delay, ASD traits, and dysmorphic appearance.	180 K	Normal	*KAT6A*	NM_006766.4	het	c.3768_3769del	p.Asp1256GlufsTer4	Dn
20NG0567	M	24	ID with predominance in language, epilepsy, and ASD traits.	≥60 K	Normal	*TCF4*	NM_001083962.1	het	c.1732C>G	p.Arg578Gly	n/a
19NG0182	F	14	ASD, hearing loss, and ptosis	≥60 K	Normal	*PMM2*	NM_000303.2	het	c.91T>C	p.Phe31Leu	Pat asym
						*PMM2*	NM_000303.2	het	c.368G>A	p.Arg123Gln	Mat asym
20NG0520	M	5	ID, autistic traits, and facial dysmorphias.	60 K	Normal	*ASXL3*	NM_030632.2	het	c.3039+2T>C	-	Dn
20NG0214	M	11	High capacities. Poor social skills.	60 K	Normal	*PIGG*	NM_001127178.2	hom	c.1515G>A	p.Trp505*	Trans
20NG0515	M	13	ID. ADHD.	180 K	Normal	*CEP290*	NM_025114.3	hom	c.2423A>G	p.Tyr808Cys	Trans
20NG0377	M	12	Psychomotor delay, hearing loss, and ASD traits	≥60 K	Normal	*ANKRD11*	NM_001256183.1	het	c.7407C>G	p.Tyr2469Ter	Dn
20NG0334	F	11	ASD and ID.	180 K	Normal	*SYNGAP1*	NM_006772.2	het	c.3778A>T	p.Lys1260Ter	Dn
20NG0237	M	12	Intellectual disability. Behaviour problems. Aggressiveness.	≥60 K	Normal	*CLTC*	NM_004859.3	het	c.3554_3555del	p.Glu1185ValfsTer10	Mat asym
19NG0460	M	6	ASD. Normal CI. Language delay. Epileptic encephalopathy.	180 K	Normal	*GRIN2A*	NM_001134407.2	het	c.1592C>T	p.Thr531Met	Dn
20NG0184	M	10	Psychomotor delay. No language. Stereotypes.	≥60 K	Normal	*IFIH1*	NM_022168.3	het	c.716dup	p.Met240HisfsTer4	n/a
20NG0113	M	8	Psychomotor delay and peculiar phenotype. ASD. No language.	≥60 K	Normal	*GABBR2*	NM_005458.7	het	c.493G>T	p.Asp165Tyr	Dn
20NG0082	F	6	ASD and ADHD. Facial dysmorphia.	≥60 K	Normal	*HUWE1*	NM_031407.6	het	c.7204+5G>A		Dn
20NG0080	M	14	Language disorder. Psychomotor delay.	60 K	Normal	*CACNA1A*	NM_001127221.1	het	c.1638C>G	p.Tyr546Ter	Dn
						*TCN2*	NM_000355.3	hom	c.185G>A	p.Ser62Asn	
20NG0003	F	5	ID with stereotypies. Epileptic seizures.	≥60 K	6q26del, 12p13.33dup	*MECP2*	NM_001110792.2	het	c.41_57dup	p.Arg20GlufsTer30	Dn
19NG1270	F	13	ASD. Subclinical epileptogenic activity.	≥60 K	Normal	*SHANK3*	NM_033517.1	het	c.3525delG	p.Asp1176ThrfsTer4	n/a
19NG1178	M	10	ASD. Intellectual disability, especially in language. ADHD. Short stature.	≥60K	Normal	*SOX11*	NM_003108.3	het	c.155C>A	p.Phe52Gln	No mat
19NG1126	F	5	Psychomotor delay. No language. Microcephaly, seizures. Peculiar phenotype.	180 K	Normal	*FOXG1*	NM_005249.4	het	c.553A>G	p.Ser185Gly	Dn
19NG1096	M	19	Major conduct disorder. Encephalopathy. Epilepsy.	60 K	Normal	*SETD1B*	NM_015048.1	het	c.4906A>G	p.Thr1636Ala	No mat
19NG1081	F	7	ID and ASD. No other malformations or distinctive features.	≥60 K	8q24.3dup, 19p12dup, Xp22.23dup, 2p22.3del	*WASF1*	NM_001024936.1	het	c.1516C>T	p.Arg506Ter	Dn
19NG1029	F	5	Psychomotor delay, ADHD, and macrocephaly	60 K	Normal	*ANKRD11*	NM_001256183.1	het	c.7083delC	p.Thr2362ProfsTer38	No mat
19NG0950	M	5	ASD with ID, no language, and macrocephaly	180 K	Normal	*KMT2E*	NM_018682.3	het	c.71+1G>T	-	Dn
19NG0781	M	6	Psychomotor delay and ASD. Dysgenesis corpus callosum. Dental alterations and macrocephaly.	≥60 K	Normal	*HIST1H1E*	NM_005321.2	het	c.446_447insT	p.Lys149AspfsTer46	Dn
19NG1312	F	11	ID. Low set hair, epicantus, anteverted nostrils, low set ears. Hearing loss.	60 K	Normal	*SETD5*	NM_001080517.2	het	c.2003C>G	p.Ser668Ter	Dn
20NG0006	M	13	ASD with ID. Peculiar phenotype. Myopia.	60 K	chr22q13.2 (42,264,616–44,068,185)X1	*NAGA, CYB5R3 TCF20*		het	Deletion	-	Dn
19NG0502	F	13	Language delay and learning difficulties. Short stature and bulbous nose.	np	np	*SRCAP*	NM_006662.2	het	c.7300G>T	p.Glu2434Ter	Dn

ASD: autism spectrum disorder; CMA = chromosomal microarray; Dn: de novo; ID: identification; Pat asym: asymptomatic father; Mat asym: asymptomatic mother; Pat sym: symptomatic father; No mat: no maternal; No pat: no paternal; n/a: not available; M: male; F: female; Het: heterozygous; Hem: hemizygous; Hom: homozygous. ID: intellectual disability; ADHD: attention deficit hyperactivity disorder; CI: intellectual coefficient; np: not performed; WES: Whole exome sequencing.

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
