# Peer review of "Towards a Change in the Diagnostic Algorithm of Autism Spectrum Disorders: Evidence Supporting Whole Exome Sequencing as a First-Tier Test"

_genes, 2021, doi:10.3390/genes12040560_

Round 1

Reviewer 1 Report

The study entitled “Towards a change in the diagnostic algorithm of Autism Spectrum Disorders: Evidence supporting whole exome sequencing as a first-tier test” aimed to study retrospectively the diagnostic utility of different genetic tests, such as FMR1 analyses, CMA and WES in patients with ASD.

The study sample (343 patients, 75% males, with age ranging from 1 to 45 years) included 4 different groups according to genetic testing performed. In the largest groups (106 patients), all study genetic analyses had been performed. Moreover, patients were classified based on their age at the time of first genetic testing (<5 or >5). CNV vere classified as pathogenic or likely pathogenic; Indeterminate CNV when pathogenicity was not clear; Unsolved or negative was used to indicate  polymorphic or unreported CNV. Data Analysis for WES was based on a custom panel including 293 genes related to ASD and variants  were classified likewise CNV and according to ACMG criteria. At the end, only characterized patients (pathogenic or likely pathogenic CNV or variants) were numbered to define the diagnostic yield.

The study reports identification of causative variants in 12.8% of the sample whereas VUS were identified in 17.8%. The overall diagnostic yield of WES was significantly higher that CMA and FMR1. Most characterized patients were actually older than 5 years old.

Overall the study is interesting, methods and results are clearly written and conclusion are consistent with the reported results. However, there are different issues that should be carefully addressed by the Authors in a major revision:

  • Taking into account that the referred patients were also clinically investigated as stated, it is particularly important to describe whether a standardized assessment for ASD diagnosis was conducted according to DSM V criteria.
  • Similarly a more detailed description of possible co-occurring conditions should be reported. In particular, the number of patients with co-occuring intellectual disability should be described.
  • In the characterized patients, it is particularly relevant to add some discussion about previously reported association of the involved genes with ASD. Some times, this is easily understandable as for example for PMM2 mutation. In other patients, this is not the case. For example, one of the characterized patients had PMM2 variants, however, PMM2 pathogenic variants cause PMM2-CDG. This is the most common among congenital disorders of glycosylation and it is not associated with ASD although some patients with profound intellectual disability might have some ASD-like symptoms (i.e. hand stereotypes, language absence).
  • The term maturational delay is unclear and should be avoided. Please describe the presence of developmental delay or intellectual disability in the study patients.
  • It is important to report on possible clinical correlates previously associated with causative CNV in patients with ASD (i.e. Int J Dev Neurosci. 2020 Jun;80(4):276-286. doi: 10.1002/jdn.10024.

Author Response

Reviewer #1

1.Taking into account that the referred patients were also clinically investigated as stated, it is particularly important to describe whether a standardized assessment for ASD diagnosis was conducted according to DSM V criteria.

We completely agree with this observation. Patients included in the study come from three different hospitals: Hospital Universitario 12 de Octubre, the Hospital Universitario de Getafe or the Hospital Universitario de Mostoles. However, all patients under clinical examination and were diagnosed according to DSM-5 clinical criteria.

The following sentence is now included (Lines 88-90):

“ASD diagnosis was performed according to DSM-5 clinical criteria. All the patients showed persistent deficits in social communication and interaction as well as restricted, repetitive behaviours”.

2. Similarly a more detailed description of possible co-occurring conditions should be reported. In particular, the number of patients with co-occurring intellectual disability should be described.

Thanks for this observation.

A brief description of the heterogeneous co-occurring conditions referred in our cohort of ASD patients is now included in the Methods section. Intellectual disability was the most frequent and it is now indicated in the manuscript.

Lines 94-99. Lines 343-344

3. In the characterized patients, it is particularly relevant to add some discussion about previously reported association of the involved genes with ASD. Sometimes, this is easily understandable as for example for PMM2 mutation. In other patients, this is not the case. For example, one of the characterized patients had PMM2 variants, however, PMM2 pathogenic variants cause PMM2-CDG. This is the most common among congenital disorders of glycosylation and it is not associated with ASD although some patients with profound intellectual disability might have some ASD-like symptoms (i.e. hand stereotypes, language absence).

The main goal of our manuscript was to describe and compare the clinical utility of the different diagnostic genetic tests used in the clinical practice. The requested information is very interesting but we believe too detailed information on that would take away the objective of the study. Therefore, as it has been purposed, we have added some little and general information of the involved genes and its association with ASD, referring the SFARI database where the majority of the involved genes are included.

Lines 306-309

4. The term maturational delay is unclear and should be avoided. Please describe the presence of developmental delay or intellectual disability in the study patients.

Thank you for this appreciation. The term maturation delay has been modified where appropriate.

Lines 221, 343, table 3 and 4

It is important to report on possible clinical correlates previously associated with causative CNV in patients with ASD (i.e. Int J Dev Neurosci. 2020 Jun;80(4):276-286. doi: 10.1002/jdn.10024.

We agree it is an interesting point. This issue is now discussed in the manuscript. The article has also been added in references.

Lines 347-352

Reviewer 2 Report

This paper is interesting and I think very timely. It is a small(ish) study in the grand scheme of things, but likely representative of more broad questions. There can be concerns that these detection methods by themselves don't actually prove autism causation, but that is a somewhat pedantic argument that could be tightened up with better wording. I actually agree with the conclusions that the authors draw, but I think that some important data are missing or not obvious.

The argument the authors are making seems to be that WES should supplant CMA. Fair enough. To prove this, I think that they need to show that all the cases detected by CMA would also be detected by WES (or at least the majority of them), and that there are cases detected by WES that are not detected by CMA. Essentially, of the 189 cases that received both CMA and WES, what were the results. While it is interesting to generally know what % of positive were detected by CMA and WES separately, this doesn't tell us if one is better than the other. 

Additionally there are other concerns that affect first-tier testing applicability. Specifically, the authors should address costs, turnaround time, and other practical concerns. The FMR1 testing, for example, may characterize relatively few cases, but is so cheap and easy that it may still be worth doing. Certainly CMA and WES are much more similar than either are to the FMR1 testing. 

Overall, I think that this is a meaningful study and I agree with the authors' conclusions, but I think that the data they show need to be presented so as to facilitate a direct comparison between the methods.

Author Response

Reviewer #2

1. The argument the authors are making seems to be that WES should supplant CMA. Fair enough. To prove this, I think that they need to show that all the cases detected by CMA would also be detected by WES (or at least the majority of them), and that there are cases detected by WES that are not detected by CMA. Essentially, of the 189 cases that received both CMA and WES, what were the results. While it is interesting to generally know what % of positive were detected by CMA and WES separately, this doesn't tell us if one is better than the other.

We completely agree with this comment.

We believe all the cases detected by CMA would have been detected by WES, as the genes encompassing the CNVs identified by CMA are included in the panels and and WES included enough probes within all CNV to possible detected them.

We have improved the sentence in the Discussion on that so it is clearer.

Lines 327-329

2. Additionally there are other concerns that affect first-tier testing applicability. Specifically, the authors should address costs, turnaround time, and other practical concerns. The FMR1 testing, for example, may characterize relatively few cases, but is so cheap and easy that it may still be worth doing. Certainly CMA and WES are much more similar than either are to the FMR1 testing. 

Effectively, this is an important point to consider.

As mentioned, the turnaround time and cost of WES and CMA are similar and, taking into account all the mentioned advantages of WES when comparing to CMA, the price should not be a priority when deciding WES as a first-tier testing. Lines 320-321

Unfortunately, we have not detail assessed cost-effectiveness in this study. This is now mentioned when describing the limitations of our study (Lines 369-370)

We agree that FMR1 testing is cheap and easy to perform and therefore, it should not always be replaced by WES. However, it is true that based on a recent paper and on our own experience, the yield of FMR1 testing is very low and in the majority of cases there is family history or clinical suspicion (Lines 286-288). Therefore, even though it is easy and cheap, performing FXS in all case would consume time and money unnecessarily. Anyway, as it was indicated in lines 77 and 78, FMR1 testing still may be performed, especially if there is clinical or familial suspicion. To clarify this, it has also been included in the Abstract (lines 40-41).

Round 2

Reviewer 1 Report

The article has been significantly ameliorated and it is fine to me.

Reviewer 2 Report

The authors have addressed my concerns.